# A neurotoxin that specifically targets *Anopheles* mosquitoes

Estefania Contreras[1], Geoffrey Masuyer [2], Nadia Qureshi[1], Swati Chawla[1], Harpal S. Dhillon[1], Han Lim Lee [3], Jianwu Chen[1], Pål Stenmark[2,4] & Sarjeet S. Gill [1]

Clostridial neurotoxins, including tetanus and botulinum neurotoxins, generally target vertebrates. We show here that this family of toxins has a much broader host spectrum, by identifying PMP1, a clostridial-like neurotoxin that selectively targets anopheline mosquitoes. Isolation of PMP1 from *Paraclostridium bifermentans* strains collected in anopheline endemic areas on two continents indicates it is widely distributed. The toxin likely evolved from an ancestral form that targets the nervous system of similar organisms, using a common mechanism that disrupts SNARE-mediated exocytosis. It cleaves the mosquito syntaxin and employs a unique receptor recognition strategy. Our research has an important impact on the study of the evolution of clostridial neurotoxins and provides the basis for the use of *P. bifermentans* strains and PMP1 as innovative, environmentally friendly approaches to reduce malaria through anopheline control.

[1] Department of Molecular, Cell and Systems Biology, University of California, Riverside, CA 92521, USA. [2] Department of Biochemistry and Biophysics, Stockholm University, 106 91 Stockholm, Sweden. [3] Unit of Medical Entomology, Institute for Medical Research, Jalan Pahang, 50588 Kuala Lumpur, Malaysia. [4] Department of Experimental Medical Science, Lund University, Lund 22100, Sweden. Correspondence and requests for materials should be addressed to P.S. (email: stenmark@dbb.su.se) or to S.S.G. (email: sarjeet.gill@ucr.edu)

Malaria continues to impact populations with significant deaths worldwide. Various approaches used to attenuate this disease's incidence include the control of *Anopheles* mosquito species that vector plasmodium transmission. Presently, chemical insecticides are the main stay of such vector control programs, although the development of resistance has impacted their use. Biological approaches used in vector control can be effective, as evidenced by successful intervention of *Onchocerca* parasitic worm transmission in West Africa using *Bacillus thuringiensis israelensis* (*Bti*)[1]. In its over two-decade use controlling *Simulium* blackflies, no resistance was observed to *Bti*, but this biological is not effective against anophelines.

With the aim to identify additional biologicals, *Paraclostridium bifermentans* subsp. *malaysia* (*Pbm*) was isolated from a mangrove swamp soil in Malaysia and this strain has high mosquitocidal activity primarily to *Anopheles*[2], even though the *P. bifermentans* (*Pb*) type strain is not mosquitocidal. *Pbm* is innocuous to mammals, fish, and non-target invertebrates[3]. Unfortunately, the lack of knowledge about the toxins involved in the mosquitocidal activity prevents its utilization as a bioinsecticide.

Previous attempts to characterize *Pbm* toxic components led to the identification of Cry16 and Cry17, which have similarity to *B. thuringiensis* Cry toxins and two proteins with low amino acid similarity to *Aspergillus fumigatus* hemolysins[4]. These proteins, as a complex, are orally toxic to *Aedes* mosquito larvae but not to *Anopheles*, even though the *Pbm* strain is more toxic to *Anopheles*[5].

Here we analyze by comparative genomics *Pb* mosquitocidal and non-mosquitocidal strains and show that the active component in *Pbm* responsible for *Anopheles* toxicity is a complex that contains a clostridial neurotoxin (CNT) that has high selectivity to anopheline mosquitoes, acquired through a megaplasmid.

## Results

**Genome sequencing of *C. bifermentans* strains**. To identify mosquitocidal components, we sequenced genomes of two *Pb* mosquitocidal strains *Pbm* and *Pb paraiba* (*Pbp*)[6], which show higher selectivity to *Anopheles* than *Aedes* mosquitoes (Supplementary Table 1), and the non-mosquitocidal *Pb*, *Pbm*, *Pbp*, and *Pb* genomes have similar chromosome sizes and belong to the group of extremely low GC clostridia, with 28% content (Supplementary Table 2, Supplementary Fig. 1A).

Eight extra scaffolds from *Pbm* sequencing data did not match chromosomic sequences. PCR amplification from these scaffolds' ends confirmed their circularity, and identified them as the *Pbm* plasmids. Similarly, PCR confirmed the presence of five *Pbp* and two *Pb* plasmids. Notably, the mosquitocidal strains share four plasmids, which were not present in non-mosquitocidal *Pb* (Table 1).

**_Pbm_ toxicity is linked to a plasmid with two toxin loci**. Loss of function *Pbm* mutants were generated by γ-irradiation. A mutant, *Pbm*Δ109, which completely lost toxicity against *Aedes* and *Anopheles* larvae, was sequenced. Data showed that this non-toxic mutant lost four *Pbm* plasmids, which are also present in *Pbp* (Table 1). A 109 kb plasmid was analyzed (Fig. 1a, Supplementary Data 1) and contained *cry16A/17A* and hemolysin-like genes in a *cry* operon, previously characterized[4,5]. We found a second toxin locus (*ptox*) (Fig. 1a) flanked by insertion sequences and transposon elements. This locus encodes a protein, which we named paraclostridial mosquitocidal protein 1 (PMP1), with high similarity to CNTs, a group that includes the tetanus neurotoxin (TeNT) and botulinum neurotoxins (BoNTs). Recently, similar putative BoNT genes were reported in non-clostridial species

from *Weissella* (BoNT-Wo) and *Enterococcus* (BoNT-En)[7–9]. The *ptox* locus has genes, which encode for non-toxic non-hemagglutinin (NTNH), OrfX1, OrfX2, OrfX3, PMP1, and P47 proteins, and a putative metallophosphatase family protein (MPP) (Fig. 1b).

BoNTs characterized to date mainly affect mammals and avian populations to various degrees, with BoNTs A, B, and E the main agents of human botulism, whereas BoNTs C and D are preponderant in cattle[10]. BoNTs intoxicate mainly by ingestion, thus they resist extreme pH and gut proteolysis to reach the bloodstream and then the nerve terminals[11]. Following receptor binding, the toxin is endocytosed and the acidic vesicular pH causes a conformational change that mediates translocation of the light chain (LC) within the neuron cytosol where LC, a zinc endopeptidase, cleaves its target SNARE proteins[12]. In the gut, BoNTs travel as high-molecular-weight complexes with associated protein components, like NTNH and the HA proteins, which stabilize the toxin[13] and promotes crossing of the host intestinal barrier[14–16]. The function of the OrfX proteins remains unknown; however, recent structural information suggests they may be involved in lipid interactions[17,18].

*Pbm* NTNH, OrfX1–3, PMP1, and P47 proteins have 35–57% amino acid identity to *Clostridium* proteins. PMP1's closest relative is BoNT/X from *C. botulinum* strain 111 (36% identity)[19], followed by BoNT/En, the *Enterococcus* BoNT-like protein[7,9] (34% identity) (Fig. 1c, Supplementary Fig. 1B). PMP1 presents the conserved SxWY motif in the binding domain ($H_C$), which in BoNTs is involved in ganglioside receptor binding (Fig. 1d, Supplementary Fig. 1C), as well as the conserved disulfide bond that links the toxin heavy and light chains, and is essential for toxicity[20]. The zinc-coordinating motif HExxH, which confers the LC its metalloprotease activity is also conserved (Fig. 1d, Supplementary Fig. 1D).

The *ptox* locus has a gene organization with an OrfX1–3 gene cluster located between NTNH and PMP1 under the same promoter (Fig. 1b). This configuration, which differs from other CNT loci, suggests that the horizontal gene transfer to *Pbm* or *Pbp* likely occurred from an ancestral bacterium as speculated for the *Enterococcus* BoNT-like cluster[20].

**_Pmp_ operon proteins show oral toxicity to *Anopheles* larvae**. PMP1 was immunodetected as a ~140 kDa protein in *Pbm* cultures (Supplementary Fig. 2A). High molecular complexes from *Pbm* were concentrated[21] and the sample, which contained PMP1 and Cry16A (Supplementary Fig. 2B), was separated by native PAGE, subjected to analysis by ultra-performance liquid chromatography-tandem mass spectrometer (UPLC/MS/MS), and compared with a similar extracted fraction from the *Pbm*Δ109 mutant (Supplementary Fig. 2C, 1st lane, Fig. 2a). All proteins from the *cry* and *ptox* loci were detected in the extracted *Pbm* sample (Supplementary Table 3), but as expected, absent in

**Table 1 Presence of plasmids (marked with X) in mosquitocidal and non-mosquitocidal *Pb* strains**

|  | *Pb malaysia* (toxic) | *Pb paraiba* (toxic) | *Pb* (non-toxic) | *Pbm*Δ109 (non-toxic) |
|---|---|---|---|---|
| Genome → size (kb) | ~3.9 | ~3.9 | ~3.6 | ~3.7 |
| Plasmid↓ size (kb) |  |  |  |  |
| 1.84 | X | X |  |  |
| 1.96 | X |  | X | X |
| 3.6 | X |  |  | X |
| 4 | X | X |  |  |
| 7.2 | X | X |  |  |
| 14.8 | X | X | X | X |
| 35.8 | X |  |  | X |
| 109 | X | X |  |  |

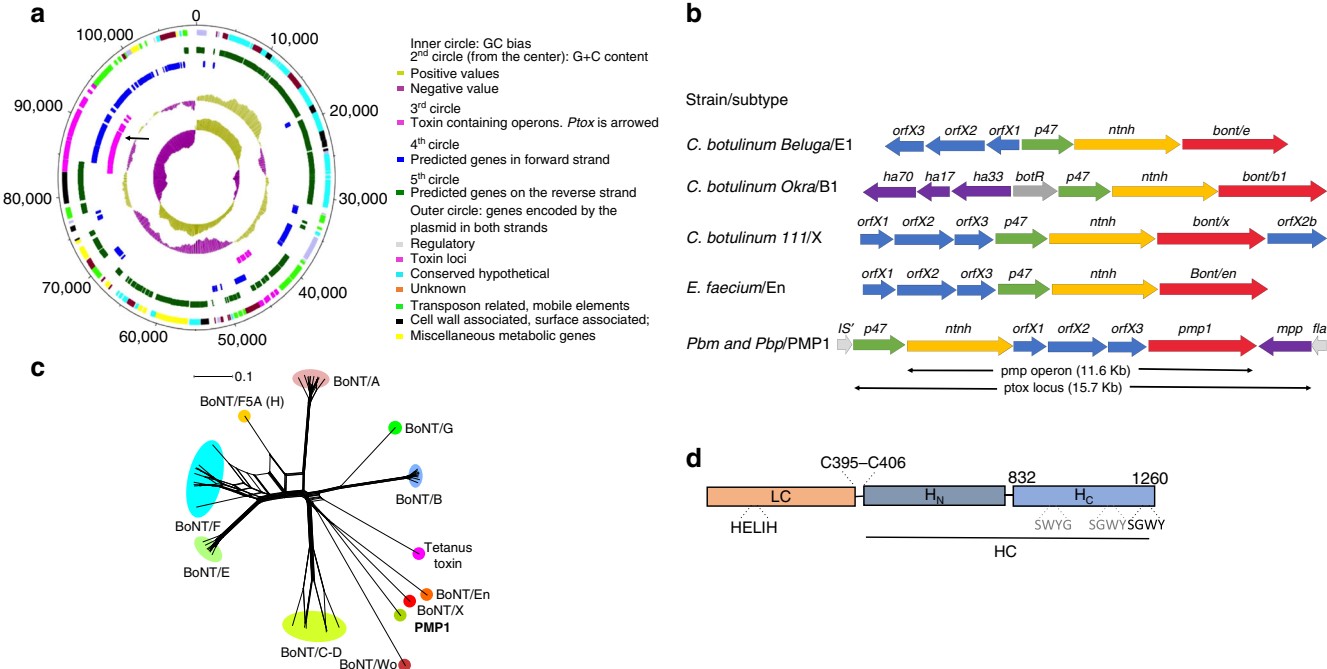

**Fig. 1** Analysis of *ptox* locus and paraclostridial mosquitocidal protein 1 (PMP1) sequences. **a** Map of the 109 kb megaplasmid in *Paraclostridium bifermentans* subsp. *malaysia* (*Pbm*) and *Pb paraiba* (*Pbp*). **b** Configuration of clostridial neurotoxin loci in different *Clostridium* and non-*Clostridium* strains and in *Pbm*. **c** Phylogenetic split network covering all botulinum neurotoxin (BoNT) serotypes, subtypes, mosaic toxins, and tetanus neurotoxin (TeNT). PMP1 is in bold. The scale bar represents the split support for the edges[47]. **d** Schematic drawing of the three domains of PMP1 showing the conserved elements found in clostridial neurotoxins: HELXH motif in the light chain (LC), the cysteines that form disulfide bond between LC and heavy chain (HC) and the tandem SxWY motif in the HC

*Pbm∆109*. Since proteins from the *cry* locus are inactive against *Anopheles* mosquitoes[5], our data provide strong evidence that proteins encoded by the *ptox* locus are responsible for the larvicidal activity to *Anopheles*.

To verify that the *pmp1* operon encodes the anopheline active toxin, we expressed these proteins in different combinations in *B. thuringiensis* 4Q7 strain (Fig. 2b). The *Bt* cultures expressing either PMP1 or NTNH protein alone had no toxicity to *Anopheles coluzzi*. However, cultures expressing both NTNH and PMP1 proteins showed 33% mortality, whereas the one expressing the full operon (Supplementary Fig. 2C, 2nd lane) had 70% mortality (Fig. 2b). Hence, the NTNH protein likely protects the PMP1 toxin from degradation in the gut, a mechanism common among BoNT complexes[22]. The OrfX1–3 proteins could also act similarly and/or facilitate PMP1 toxin absorption in the gut, since increased activity is observed in their presence than in their absence (Fig. 2b). Importantly, our data show a role in toxicity for the OrfX proteins. None of the constructs was significantly toxic to *Ae. aegypti*. Thus, the selectivity observed in *Pbm* to *Aedes* is likely produced by the *cry* operon alone[5].

The toxicity of *Bt* expressing the *pmp* operon is lower than that of wild-type *Pbm*. Potentially other proteins in the megaplasmid may contribute to the anopheline toxicity. Alternatively, differences in toxin stability and/or differences in toxin availability in *Pbm* and *Bt* contribute to the lower toxicity of the latter.

**PMP1 is toxic to mosquito larvae in vivo**. To evaluate if PMP1 alone is toxic when the gut barrier is by passed, we injected recombinant PMP1 into mosquito larvae (Fig. 2c). Strikingly, injected PMP1 was toxic to both *Aedes* and *Anopheles* mosquitoes with an $LD_{50}$ of 14 pg (98 amol) and 6.5 pg (44.5 amol) per larva, respectively. The toxicity observed is 10–100 times greater than that of spider toxins that were also injected[23].

*Aedes* larvae injected with the $LC_{99}$ (54 pg/larva) fully recovered from the injection, but at 3 h showed significant slowing of motion (Fig. 2d, Supplementary Movie 1), consistent with the paralysis associated with CNTs' intoxication. PMP1 was also toxic to adult mosquitoes by injection, since a dose-dependent impairment in their ability to fly was observed (Fig. 2e). Pre-incubation of the toxin with the metalloprotease inhibitor 1,10-phenanthroline before injection decreased PMP1 toxicity (Fig. 2f). Further, the mutation E209Q in the metalloprotease active site (HExxH motif) abolished activity (Fig. 2c), which confirms that PMP1 is a metalloprotease and this activity is essential for toxicity.

Since PMP1 is toxic to both mosquito species by injection, we determined if its specificity extends to other diptera or mammals once the gut barrier is bypassed. Although *Pbm* culture was not toxic by feeding to *Drosophila* larvae and adults (Supplementary Table 1), recombinant PMP1 was toxic to adult flies by injection (Fig. 2e). Further, PMP1 shows no toxicity to mice by the Digit Abduction Score assay and by intraperitoneal injections.

**PMP1 cleaves mosquito syntaxin**. LC metalloprotease activity is specific for one of the three neuronal SNARE proteins in mammals and their cleavage prevents neuro-exocytosis. In particular, VAMP-2/*n*-synaptobrevin is the target of TeNT and BoNTs B, D, F, G, X[19,24–26], BoNT/Wo, and BoNT/En[9,27], while syntaxin 1 is the target of BoNT/C and SNAP-25 is cleaved by BoNTs A, C, and E and BoNT/En[9,28,29]. To determine if PMP1 can cleave one of these SNARE protein homologs in mosquitoes, we incubated recombinant *Anopheles gambiae* syntaxin1A, *n*-synaptobrevin, and SNAP-25 with PMP1 LC. Only the C terminus of mosquito syntaxin was cleaved by PMP1 LC but not by the catalytically inactive PMP1 LC E209Q mutant (Fig. 3a). Interestingly, PMP1 LC was unable to cleave the recombinant human syntaxin1A

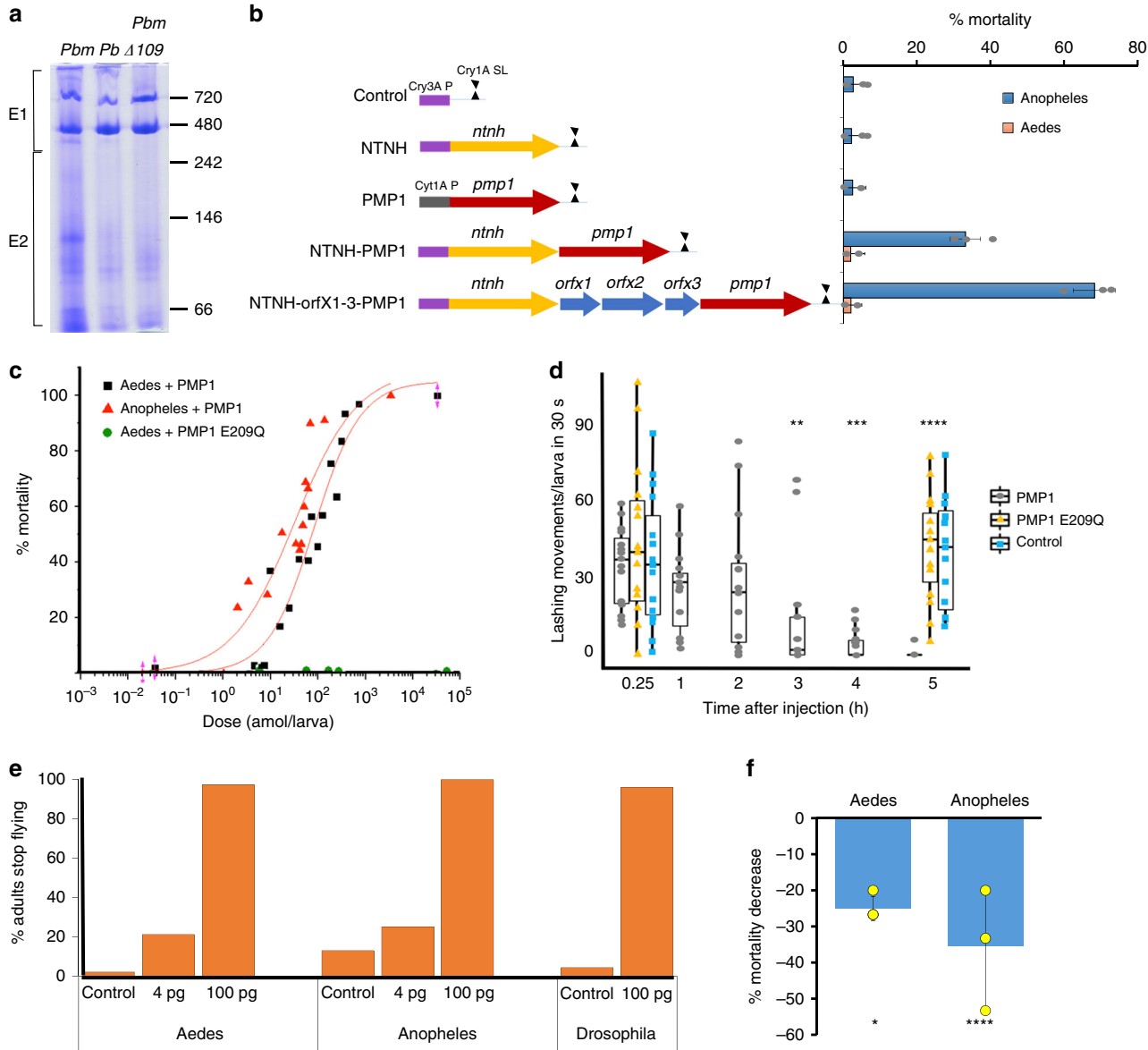

**Fig. 2** Paraclostridial mosquitocidal protein 1 (PMP1) and its associated proteins are toxic to mosquitoes. **a** Native PAGE of *Paraclostridium bifermentans* subsp. *malaysia* (*Pbm*), *P. bifermentans* (*Pb*), and *PbmΔ109* extracted fractions. Lanes were split into E1 and E2 for tandem mass spectrometry (MS/MS) analyses. Greater abundance of OrfX1–3 in E2 may indicate that they are not strongly associated to a high molecular weight complex. **b** Schematic of constructs expressing proteins in the *pmp* operon (left panel) and their corresponding mortality to *Anopheles* and *Aedes* mosquito larvae (right panel) (Supplementary Note 1) (*n* = 15/assay, see Methods). The lethal concentration 50 (LC50) values of the *pmp* operon is 0.42 and 0.48 ml in two replicates. **c** Toxicity of PMP1 and inactive PMP1 E209Q mutant to larvae by injection in a dose–mortality plot. **d** *Aedes* 3rd instar larvae (*n* = 15) show reduced movement after injection with PMP1, but not when injected with water or inactive PMP1 E209Q mutant (*t* test, *p* value ≤0.05). Median in bold, box edges are 25th and 75th percentiles and vertical lines are min and max values. **e** Adult mosquitoes and flies (percent) stopped flying after 24 h of injection with PMP1. **f** Pre-incubation with 1,10-phenanthroline decreases PMP1 toxicity by injection (*t* test, *p* value ≤0.05). Error bars represent ± s.d. of three replicates. Source data are provided as a Source Data file

(Fig. 3a), the C terminus of which is identical in mouse, and hence consistent with the lack of toxicity of PMP1 to mice.

To determine the PMP1 cleavage site, a peptide of ~4.5 kDa released from syntaxin cleavage (Fig. 3b) was purified and analyzed by UPLC-MS/MS and the peptide HAMDYVQ-TATQDTKK was detected (Supplementary Fig. 3B). Since *Anopheles* syntaxin1A C terminus is rich in positive charges, making it difficult for MS/MS detection, a syntaxin mutant (*syxΔ2myc*), where the charged region was deleted and an additional myc tag added, was created and similarly analyzed (Supplementary Fig. 3A). MS/MS identified peptides from the C terminus of syntaxin from H255 (Supplementary Fig. 3B). Thus,

PMP1 LC cleaves syntaxin between E254 and H255 releasing a peptide that matches the observed size (Fig. 3b).

PMP1 does not cleave human syntaxin despite the cleavage site being conserved (Fig. 3c). However, a region closer to the C terminus shows sequence variation between human and mosquito syntaxins that could potentially influence substrate recognition. Indeed, CNTs are known to have an extensive substrate length requirement that involve multiple exosites away from the catalytic pocket[30]. To test this hypothesis, we generated *Anopheles* syntaxin single or double mutants where we changed amino acids in this region to the corresponding residues in human syntaxin (Fig. 3c) and incubated with PMP1 LC. These mutants

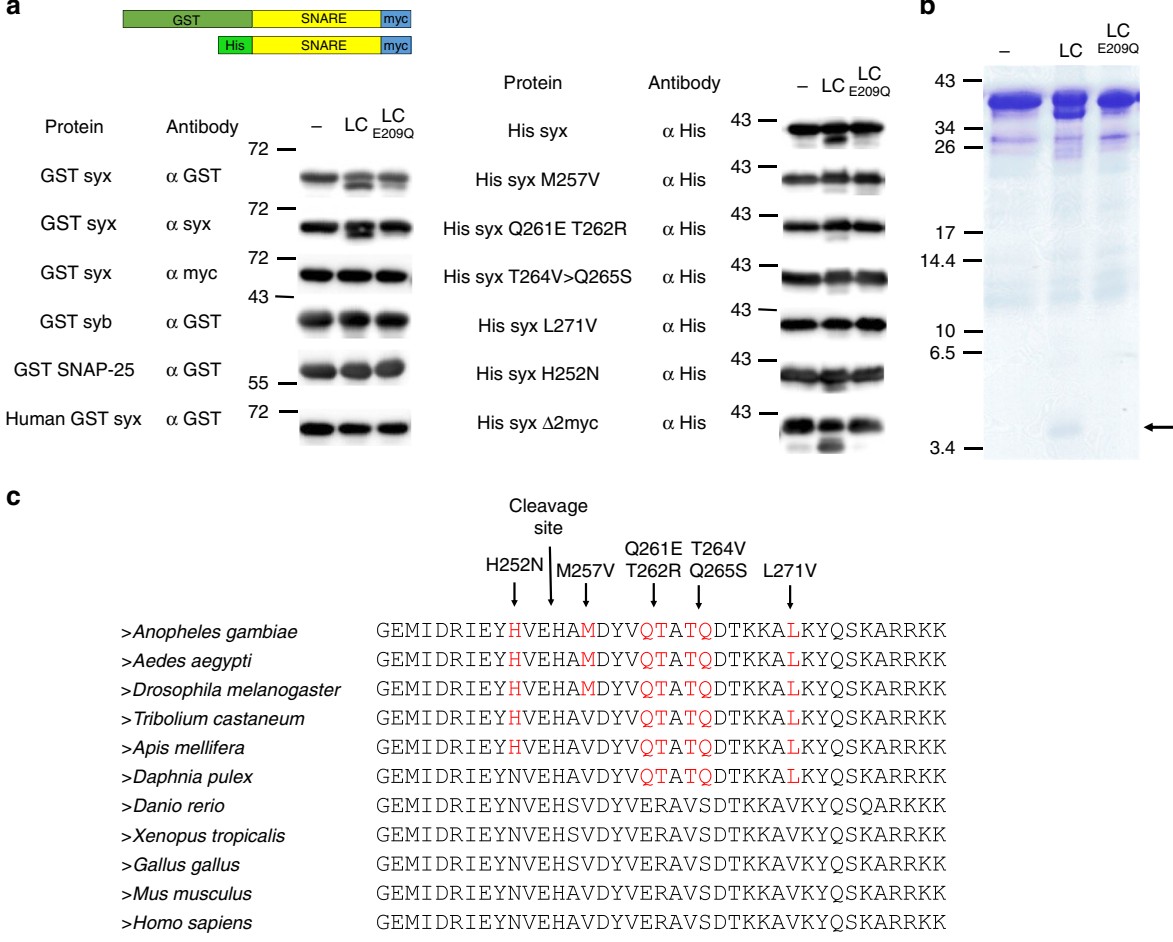

**Fig. 3** Paraclostridial mosquitocidal protein 1 (PMP1) light chain (LC) cleaves mosquito syntaxin. **a** Representation of the recombinant tagged SNARE proteins used in PMP1 LC cleavage assays (upper panel). Immunodetection of SNARE proteins and syntaxin mutants performed in the absence or in the presence of PMP1 LC, and PMP1 catalytically inactives E209Q mutant. **b** Sodium dodecyl sulfate-polyacrylamide gel electrophoresis (SDS-PAGE) of His-syntaxin cleavage assay. The fragment of 4.5 kDa band released after incubation with PMP1 LC is indicated and was analyzed by ultra-performance liquid chromatography-tandem mass spectrometer (UPLC/MS/MS). **c** Alignment of the C-termini of syntaxins in different species. Amino acids that differ from human syntaxin are in red. The position of PMP1 cleavage site and the mutations introduced in syntaxin and tested in cleavage assays are indicated with arrows. Source data are provided as a Source Data file

cleaved less efficiently than *Anopheles* syntaxin, and L271V completely abolished cleavage (Fig. 3a). The presence of non-polar amino acids, such as valine in human syntaxin C terminus, likely disrupts the interaction with PMP1 LC and prevents cleavage.

**Canonical CNT GBS is not critical for PMP1 mosquito toxicity.** CNTs bind to the presynaptic membrane of peripheral nerve terminals with high specificity[31]. Polysialogangliosides recruit and concentrate CNTs, while specific proteins such as associated vesicle protein 2 (SV2) or synaptotagmin have been described as cell entry mediators[12]. A conserved ganglioside-binding site (GBS), which contains the SxWY motif has been identified in the C terminus of BoNT/A, B, E, F, G, X and TeNT $H_C$, where the tryptophan is essential in maintaining hydrophobic interactions with the ganglioside sugar and is involved in toxicity[32]. Notably, PMP1 also shares a SGWY motif in the corresponding site, and a tandem repeat in front (Fig. 1d, Supplementary Fig. 1C).

To analyze if the SxWY motif is involved in toxicity and to determine the presence of other cell recognition regions, recombinant PMP1 $H_C$ was crystallized (residues 825–1260) and its structure determined at 1.95 Å resolution (Table 2). PMP1 $H_C$ has a fold similar to other CNTs[33] and a well-conserved

secondary structure (Fig. 4a, Supplementary Fig. 4A, B) despite having <30% sequence identity. PMP1 $H_C$ includes two subdomains, a N-terminal lectin-like fold ($H_{CN}$) consisting primarily of 15 β-strands arranged in a jelly-roll fold and a C-terminal β-trefoil fold mostly composed of seven pairs of β-strands linked by loops ($H_{CC}$) (Fig. 4a). The $H_{CC}$ is the main region associated with cell recognition, and in PMP1, presents the most structural variation compared to other CNTs. One striking feature of PMP1 $H_{CC}$ is its general hydrophobicity, with an array of a dozen aromatic residues exposed on its surface (Fig. 4b) and its lack of clear binding pockets (Supplementary Fig. 4C). The SxWY motif (S1229, W1231, and Y1232) is in a shallow pocket flanked on one side by a short lysine-rich α-helix (1242–1246) that is unique to PMP1 and may prevent the binding of gangliosides with branched carbohydrate head groups (Fig. 4b, Supplementary Fig. 4C).

Of particular interest are the three extended loops found almost in parallel in the $H_{CC}$ subdomain. Extended loops 1094–1099 (loop 1), 1164–1172 (loop 2), and 1202–1208 (loop 3) (Fig. 4b) present bulky hydrophobic side chains. Loops 1 and 2 come close to each other with E1099 making a salt bridge with K1168 (Fig. 4b), at a location where BoNT/B and TeNT bind synaptotagmin and disialyllactose or a tri-peptide, respectively[34]

**Table 2 Data collection and refinement statistics (molecular replacement)**

|  | PMP1 H$_C$ |
|---|---|
| Data collection |  |
| Space group | C2$_1$ |
| Cell dimensions |  |
| a, b, c (Å) | 118.2, 38.7, 108.5 |
| α, β, γ (°) | 90.0, 116.7, 90.0 |
| Resolution (Å) | 48.5–1.95 (2.00–1.95)[a] |
| R$_{merge}$ | 0.097 (0.73)[a] |
| I/σI | 8.3 (1.6)[a] |
| Completeness (%) | 96.9 (94.4)[a] |
| Redundancy | 3.5 (3.3)[a] |
| Refinement |  |
| Resolution (Å) | 48.5–1.95 |
| No. of reflections | 29,725 |
| R$_{work}$/R$_{free}$ | 17.8/20.9 |
| No. atoms |  |
| Protein | 3510 |
| Ligand/ion | 11 |
| Water | 257 |
| B-factors |  |
| Protein | 35.2 |
| Ligand/ion | 61.5 |
| Water | 42.1 |
| R.m.s. deviations |  |
| Bond lengths (Å) | 0.008 |
| Bond angles (°) | 1.20 |

*PMP1* paraclostridial mosquitocidal protein 1, *R.m.s.* root mean square
[a]Values within parentheses are for highest-resolution shell

(Supplementary Fig. 4C), suggesting a different cell recognition mechanism for PMP1. A fourth loop composed of residues 1213–1222 (loop 4) was disordered, but seems to occupy a different position compared to the equivalent loops in BoNT/B and TeNT that are more protuberant. This loop was recently shown to play an essential role in lipid bilayer insertion for BoNT/B, DC, and G[19,35].

The toxicity of PMP1 H$_C$ mutants was tested by injecting an LC$_{99}$ dose into *A. aegypti* larvae (Fig. 4c). Remarkably, W1231A and W1224A mutations of the tandem SxWY motifs did not decrease toxicity (Fig. 4c). The mutation of S1229, which disrupts ganglioside binding in BoNTs[36], did not affect toxicity either (Fig. 4c). Mutation of an exposed F1202 in loop 3 did not decrease toxicity, but other mutations on loops 1 to 3 decreased toxicity to different extents. We observed a slight reduction in the quadruple mutants of loops 2 and 3 (K1168A/I1169A/K1170A/E1171A and M1165A/Y1166D/M1204A/Y1205D) and a strong reduction to 8% mortality in the quadruple mutant of loop 1 (S1095A/W1096A/Y1097A/G1098A) (Fig. 4c). Interestingly, mutation of Trp alone (W1096A) in loop 1 strongly decreased the toxicity to 30%. In addition, the mutant affecting a surface-exposed tyrosine hydrophobic patch (Y1100D/Y1101D/Y1173A/Y1227A) also nearly abolished toxicity (Fig. 4c, Supplementary Fig. 4).

Our data suggest that the canonical SxWY motif included in the GBS is not essential for toxicity. Instead, a similar motif, SWYG, located in loop 1 of the H$_C$ C terminus is critical, with an important role for the tryptophan in this loop.

## Discussion
Our data support an event of plasmid acquisition by the non-mosquitocidal *Pb*-type strain. The mosquitocidal strains *Pbm* and

*Pbp* share a similar genome with *Pb*, but contain extra plasmids and in particular a megaplasmid with two toxin loci that provides toxicity to mosquito species. The isolation of these strains from anopheline endemic areas provides strong evidence of an example of insect–bacteria coevolution. Notably, both strains, isolated from Malaysia and Brazil, respectively, share a nearly identical megaplasmid, although originally from distant locations. Thus, it is likely that similar bacterial strains can be isolated from other locations populated with *Anopheles* species, facilitating the use of native strains for *Anopheles* control programs in these areas.

Our results show for the first time that the operon-associated OrfX proteins play a role in the toxicity of a CNT. As with mammalian BoNT, the *Pbm* NTNH increases the toxicity of PMP1. Oral toxicity appeared specific to *Anopheles* mosquitoes, and might be linked to the inability of PMP1 to cross the midgut barrier in *Aedes* and *Drosophila*. It is possible that the OrfX cluster, encoded in the *pmp* operon, aids PMP1 absorption through the *Anopheles* larvae midgut, but not in other insect species tested.

PMP1 exerts its action through its metalloprotease activity on a SNARE protein in a typical BoNT-like fashion, as demonstrated by mutagenesis of the catalytic site and decreased toxicity observed in the presence of a metalloprotease inhibitor. Additionally, the mutation E209Q abolished PMP1's ability to cleave *Anopheles* syntaxin1A, a highly conserved neuronal protein. Key differences in the sequence of human/mouse syntaxin prevent its cleavage by PMP1. Our data shows that residues, which are key for PMP1 LC cleavage, are shared by dipteran species, including *Drosophila*, and some are shared among invertebrates but are absent in vertebrates, mammals, and fish (Fig. 3c). Whether PMP1 is toxic to other invertebrates by injection besides mosquitoes and *Drosophila* needs investigation. Nevertheless, the toxicity of injected PMP1 to the species tested matches the ability of PMP1 LC to cleave syntaxin. However, at this stage we cannot exclude the possibility that PMP1 cleaves other proteins and that the lack of mouse toxicity was due to the inability of PMP1 to bind or cross the mammalian neuronal membrane. Further research about the interaction between the toxin and the nervous system cell membranes in different species is needed to fully elucidate the mechanism of PMP1 internalization.

The structure of PMP1 H$_C$ shows significant differences in comparison to the structures of other CNTs. PMP1 is the first example of a CNT where the SxWY motif, although conserved, is not implicated in toxicity. Our data propose that the unique PMP1 H$_C$ hydrophobicity and a SWYG motif in loop 1 are important determinants of toxicity. In BoNT/C the SxWY motif is not conserved and one of the two GBS is located on an exposed loop[37]. It was proposed that some BoNTs interact directly with lipid membranes via hydrophobic interactions through an extended loop (loop 4)[19]. PMP1 also has hydrophobic domains that could facilitate similar interactions. However, it is possible that loop 1 (SWYG) interacts with specific mosquito midgut receptors akin to that observed with the specificity observed with insecticidal *B. thuringiensis* toxins[38].

The toxicity of recombinant BoNT/X from *C. botulinum* strain 111 to mice by injection is very low[19]. The *Enterococcus* BoNT-like protein shows no toxicity to mice, although it cleaves human VAMP-2 and SNAP-25 in neurons. BoNT/X and BoNT/En represent two members of a lineage of BoNTs[7,9] the specificities of which are unknown (Fig. 1b). Here we reveal that PMP1 is a member of this lineage and targets *Anopheles* mosquito larvae. PMP1 is thus the first CNT that targets invertebrates. Hence, clostridial-like toxins with low mammalian toxicity have organismal targets that are yet unidentified and focusing only on mammalian or vertebrate targets could be misleading. It is likely these toxins co-evolved with their target host, as observed here for PMP1 with anopheline mosquitoes.

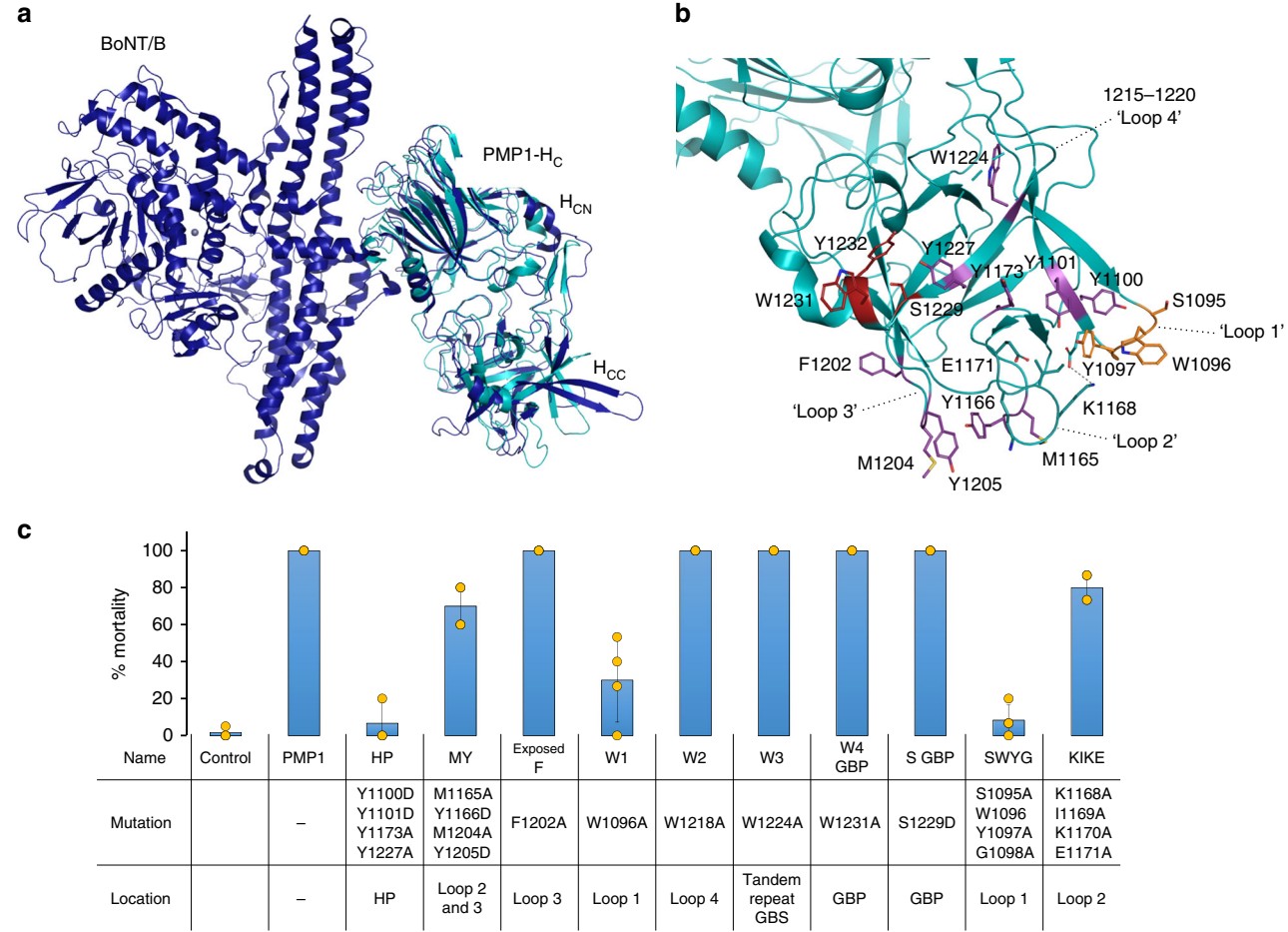

**Fig. 4** Difference in the $H_{CC}$ domain of paraclostridial mosquitocidal protein 1 (PMP1) contributes to its selective toxicity. **a** Superimposition of the PMP1 $H_C$ crystal structure (cyan) with the full-length botulinum neurotoxin (BoNT)/B (blue, PDB 1EPW). The domains are labeled as LC (catalytic light chain), $H_N$ (translocation domain), and the $H_C$ (binding domain) with its two subdomains ($H_{CN}$ and $H_{CC}$). **b** Crystal structure of PMP1 $H_C$. The aromatic residues exposed on PMP1 $H_{CC}$ surface, the mutations tested for toxicity (purple sticks), the loop 1 SWYG motif (orange sticks), and the conserved ganglioside-binding motif (red sticks) are labeled. The E1099-K1168 salt bridge is indicated as a dashed line. **c** Toxicity of PMP1 $H_{CC}$ mutants to *A. aegypti* 4th instar larvae by injection (*t* test, *p* value ≤0.05). Error bars represent ± s.d. of at least three replicates. Source data are provided as a Source Data file

We believe our contribution has very important implications about the evolutionary origin of CNTs. Broadening the range of CNTs specificity found in *Clostridium* and non-*Clostridium* species will be useful, not only for medical applications but also for novel biotechnological purposes.

As an example, *Pbm* could be formulated together with other mosquitocidal strains or the *Pbm* toxin genes incorporated into another strain. Both approaches would increase the spectrum of activity of currently used biological insecticides. However, a more attractive alternative could be to use locally isolated *Pb* mosquitocidal strains. As *Pb* is an anaerobe such mosquitocidal cultures can be readily made in rural areas with minimal inputs. These crude cultures can then be used for suppression of larval anophelines.

We open a field for targeting pests, pathogens, or vectors of diseases with a mode of action that might delay the appearance of resistance in an environment-friendly system.

## Methods

**Insects**. *Anopheles coluzzi*, *Anopheles stephensi*, and *Ae. aegypti* mosquito larvae were reared at 28 °C with a photoperiod of 16:8 h light/darkness in distilled water and fed with 1:4 yeast/fish food.

**Bacterial strains and culture conditions**. *Paraclostridium bifermentans* (ATCC 638) from the American Type Culture Collection was used as the wild-type reference strain, and *P. bifermentans* subsp. *malaysia* and *P. bifermentans* subsp.

*paraiba* were from the collection of the Institute for Medical Research, Malaysia[39]. Bacteria were grown in liquid tryptone-yeast extract-glucose (TYG) medium at 30 °C under anaerobic conditions using BD GasPakEZ (Becton-Dickinson Microbiology).

*Bacillus thuringiensis israelensis* (*Bti*) 4Q5 strain from Bacillus Genetic Stock Center was grown overnight at 30 °C in sporulation media (0.8% Nutrient broth, 1 mM $MgSO_4$, 13 mM KCl, 10 μM $MnCl_2$, 0.5 mM $CaCl_2$) with shaking until complete autolysis.

**Toxicity assays**. Different volumes of *Pbm* or *Bti* whole bacterial cultures were tested at room temperature in 100 ml water cups containing 20 third instar mosquito larvae. Bioassays were repeated at least three times.

To test the toxicity of PMP1, NTNH, NTNH-PMP1, and NTNH-OrfX1-3-PMP1 constructs, *B. thuringiensis* subsp. *israelensis* 4Q7 cells (Bacillus Stock Center, Ohio State University, Columbus, OH, USA) transformed strain was grown overnight at 30 °C in sporulation media with 50 μg/ml erythromycin and a 100× dilution in bioassay water cups was used. The experiment was repeated six times for the control construct, five times for PMP1 and NTNH-PMP1, four times for NTNH-OrfX1-3-PMP1, and twice for NTNH construct. Mortality was assessed after a 3-day exposure. To determine the $LC_{50}$ of the NTNH-OrfX1-3-PMP1 varying levels of the bacterial culture was added to 100 ml bioassay cups and the mortality was assessed after a 3-day exposure. The $LC_{50}$ were determined by probit analysis (SPSS). This determination was performed twice.

***Pb malaysia* mutagenesis**. A *Pb malaysia* overnight culture was diluted 1:30 in TYG media and grown for 6 h in anaerobic conditions. The cells were exposed to a $^{137}$Cesium source (J.L. Shepherd and Associates) for 6 min. Irradiated cells were diluted 1:100 and grown overnight at 30 °C in anaerobic conditions on TYG plates. Individual cells were selected and grown in liquid TYG for toxicity screening.

Screening of 3000 colonies was performed using three *Aedes* 2nd instar larvae in 1 ml water in 24-well polystyrene plates and toxicity was recorded after 24 h. The three mutants that lost completely their toxicity to *A. aegypti* were then bioassayed with *An. stephensi* larvae and one mutant *PbmΔ109* selected for further analysis.

**PAGE and immunoblotting**. Proteins separated in a sodium dodecyl sulfate-polyacrylamide gel electrophoresis (SDS-PAGE) or native gel were transferred onto a PVDF membrane (Immobilon P, Millipore) and incubated overnight 4 °C with the primary antibody in blocking buffer (5% skim milk in phosphate-buffered saline (PBS)). A secondary antibody, enhanced chemiluminescence anti-rabbit IgG-horseradish peroxidase-linked whole antibody (ref. NA934 GE Healthcare, Anaheim, CA, USA), was used at a 1:5000 dilution. Immunoreactive bands were visualized using the SuperSignal West Dura Extended Duration Substrate (Thermo Fisher). Rabbit antibodies against the PMP1 peptide in the heavy chain (GFE-NIDFSEPEIRY) was produced (GenScript, Piscataway, NJ, USA) and used at a 1:3000 dilution.

**Genomic DNA isolation**. For genome sequencing, total *Pb malaysia*, *Pb paraiba*, and *Pb* DNA were isolated using phenol–chloroform extraction protocol and *PbmΔ109* was isolated using DNeasy blood and tissue kit (Qiagen) from fresh overnight cultures. Quantity and quality of the DNA were measured spectrophotometrically (Nanodrop 2000, Thermo Scientific).

**Proteomic analysis**. *Pb malaysia* and *PbmΔ109* proteins present in the culture supernatant were acid precipitated adding $H_2SO_4$ dropwise to pH 3.5 as described[21]. Precipitated proteins were extracted by agitation for 2 h in 0.1 M sodium citrate buffer pH 5.5 and analyzed in native protein acrylamide gels. Protein lanes were then excised from the gel and analyzed by mass spectrometry (LTQ Orbitrap Fusion MS coupled to 2-dimension nano-UPLC) at the Proteomics Core facility at the University of California, Riverside. Protein searches were performed against *Pb malaysia* genome predicted protein database.

For analyses of the cleavage site, cleavage assay mixtures after incubation were peptide purified using Sep-Pak cartridges (Waters) and analyzed by mass spectrometry similarly.

**Larvae and adult injection**. Fourth instar larvae were kept on ice and then injected between the head and the thorax on a Petri dish using Drummond capillary tubes and a Nanoject II auto-nanoliter injector (Drummond Scientific). Injected larvae were transferred to water cups and kept for 24 h under standard rearing conditions. Adults were injected between the thorax and abdomen and transferred to adult mosquito rearing tubes.

Forty groups of 15 *A. aegypti* and 42 groups of *An. coluzzi* larvae were injected with different doses of PMP1 from 0.02 to 30,000 amol/larva. Mortality was recorded after 24 h. For replicates of the same dose, the mortality mean was used (2 replicates of 15 larvae for all doses except 97.65 and 24.53 amol/larva, which were repeated three times; 61.65 and 38.83 amol/larva were repeated five times; and 3276.15, 727.47, 15.47, 7.27, 4.52, 0.36, and 0.003 amol/larva were repeated once). Six groups of 15 larvae were injected by PMP1 E209Q. Data were adjusted to a control background mortality of 3% for *Aedes* and 7% for *Anopheles*. The dose–mortality curve was plotted using the Origin Lab software. The PMP1-injected larvae data fitted a sigmoid dose–response curve (Hill equation). For the *Aedes* curve, reduced: $\chi^2 = 79.6$ and adjusted $R^2 = 0.935$. For the *Anopheles* curve, reduced: $\chi^2 = 126.8$, adjusted $R^2 = 0.854$.

To test PMP1 toxicity to adult mosquitoes and flies, the following number of individuals were injected: 58 *Aedes* control, 60 4 pg, 43 100 pg; 54 *Anopheles* control, 65 4 pg, 62 100 pg, 15 *Drosophila* control, 15 100 pg. Mortality rate due to injection (discarded dead individuals 1 h after injection) was between 12 and 13.9% for all *Aedes* experiments, including control and independent of the dose of PMP1 injected, 42.6–50.6% for *Anopheles* and 7.4–14.8% for *Drosophila*.

To test the effect of 5 mM 1,10-phenanthroline in PMP1 toxicity, the mixture was incubated for 30 min on ice before injection. The decrease of toxicity is represented as percentage in comparison to the injection of PMP1 without inhibitor. Three replicates of 15 individuals were performed for Anopheles and 4 for Aedes.

To test the toxicity of PMP1 HC mutants by injection, three replicates of 15 individuals were performed for all constructs except MY, W1, SWYG, and KIKE constructs, which were repeated four times.

**Plasmid construction, protein expression, and purification**. The *pmp1* gene was commercially synthesized (GenScript, Piscataway, NJ, USA) using *B. thuringiensis* codon optimization. The *ntnh-orfX1-orfX2-orfX3-pmp1* genes were amplified from *Pbm* whole DNA preparation using Platinum Taq high-fidelity polymerase (Thermo Fisher) and primers 1 and 2 (Supplementary Table 4) in an automated thermocycler (C 1000 Touch, Bio-Rad). Individual *ntnh* and *pmp1* genes were amplified similarly using primers 3, 4, and 5, 6, respectively, to produce constructs NTNH and NTNH-PMP1. PCR products were separated in 1% agarose gels and subsequently cut and purified using Wizard SV gel and PCR purification kits (Promega, Madison, WI, USA). Sequencing of purified DNA products was performed by the Genomics Core facility at the University of California, Riverside. The

full operon, *ntnh*, *pmp1*, and *ntnh-pmp1*, constructs were first subcloned into pCR2.1 TOPO TA vector (Thermo Fisher) and then cloned into pHT315 vector[40] under Cyt1A promoter from *B. thuringiensis israelensis* (Cyt1A P) or Cry3A promoter from *B. thuringiensis tenebrionis* (Cry3A P) and Cry1A stem loop terminator (Cry1A SL). For expression of *pmp1* gene in NTNH-PMP1 and *orfX1*, *orfX2*, *orfX3*, and *pmp1* genes in NTNH-OrfX1-3-PMP1 construct, the native Shine–Dalgarno sequences were used. The constructs in pHT315 were transformed in *Bti* 4Q7 cells.

PMP1, PMP1 catalytically inactive E209Q mutant, PMP1 HC mutants, PMP1 LC, PMP1 $H_C$, and SNARE proteins were purified from *Escherichia coli*. PMP1 was commercially synthesized *E. coli* codon optimized and cloned in pQE-30 vector (Qiagen). Fragments of PMP1 $H_C$ containing the desired mutations were individually synthesized between restriction sites RsrII and HindIII, and were inserted in PMP1 to produce PMP1 $H_C$ mutants. Catalytically inactive E209Q mutant was created by nested PCR using primers 7, 8, 9, and 10. PMP1 $H_C$ was amplified from PMP1 gene using primers 11 and 12 and cloned in pET duet 1. PMP1 LC was amplified from PMP1 gene using primers 13 and 14 and cloned in RSF duet 1. DNA sequence encoding fragments of SNARE proteins (*A. gambiae* VAMP-2 amino acids 1–99, syntaxin 1–268, SNAP-25 1–213, and Human VAMP-2 1–93 and syntaxin 1–266) were commercially synthesized codon optimized for *E. coli* expression with a myc tag added in C terminus (GenScript, Piscataway, NJ, USA) and cloned in pGEX-6P vector. *Anopheles gambiae* syntaxin with a His tag was amplified using primers 15 and 16 (Supplementary Table 4) from synthesized syntaxin fragment and cloned in pET Duet 1. Syntaxin mutants were produced by nested PCR, inserting the desired mutations in primers 17–28 (Supplementary Table 4).

BL21(DE3) pLysS chemically competent *E. coli* cells (Agilent) were transformed with genes cloned in vectors pGEX-6P, pET Duet 1, and RSF Duet 1 according to the manufacturer's protocol. Chemically competent M15 cells (Qiagen) were used for transformation of genes cloned in pQE-30. Cells were induced by adding 1 mM isopropyl β-D-1-thiogalactopyranoside, grown in LB medium for 4 h at 25 °C, and harvested by centrifugation. Cell lysis was produced in 50 mM Tris, 300 mM NaCl, 1 mM dithiothreitol, 0.1% glycerol, 500 µg/ml lysozyme, pH 7.4, and sonicated for 3 min. PMP1 HC, PMP1, PMP1 mutants, syntaxin, and syntaxin mutants with a His tag were purified from the lysate supernatant using Ni NTA agarose beads (Qiagen). LC was purified using Flag tag affinity gel (BioLegend) and the SNARE proteins with a GST tag were purified using GST SpinTrap columns (GE Healthcare).

**Cleavage assays**. Recombinant *A. gambiae* VAMP-2/synaptobrevin, syntaxin, syntaxin mutants, SNAP-25, and human syntaxin (2 µg) were incubated in 50 mM $NaH_2PO_4$ buffer, pH 6.2, with 500 ng of LC or catalytically inactive E209Q LC for 3 h 30 °C. Samples were analyzed by SDS-PAGE and western blot and immunodetected using GST tag antibody (ref. 27457701, GE Healthcare, 1:2000 dilution), His tag antibody (ref. A00186, GenScript, 1:5000 dilution), *Drosophila* syntaxin antibody (ref. 8C3, Developmental Studies Hybridoma Bank, University of Iowa, 1:1000 dilution) or myc tag antibody (ref. 2272, Cell Signaling, 1:1500 dilution). Uncropped version of the western blots is provided in Source Data file.

**Digit abduction assay and intraperitoneal injections**. Mice, 50–55 g ($n = 3$), were anesthetized by intraperitoneal injection of ketamine/xylazine and after 15 min 1 µg of PMP1 in 100 µl PBS was injected into the gastrocnemius muscles of the mice left hind limb and the same volume of buffer was injected into the right hind limb of the same mouse as a control. Mice were monitored for any symptom of flaccid paralysis and ability to spread the toes for 24 h. Intraperitoneal injections were performed injecting 3.0 µg PMP1 in unanesthesized mice, 25–27 g ($n = 3$). Mice were monitored for mortality and visible effects for 24 h and up to 4 days. The UC Riverside animal use committee approved these protocols.

**X-ray crystallography**. Crystals of PMP1 $H_C$ (8.4 mg/ml) were obtained from a sitting-drop vapor diffusion setup against 0.12 M ethylene glycols, 0.1 M Buffer System 3, pH 8.5, 50% v/v Precipitant Mix 4 (Morpheus screen, Molecular Dimensions, UK). A drop of 200 nl of sample was mixed with an equal amount of reservoir and incubated at 21 °C. Crystals grew within 2 weeks and were frozen in liquid nitrogen for data collection. Diffraction data were collected at a wavelength of 0.984 Å at station ID30B of the ESRF synchrotron (Grenoble, France). A complete dataset at 1.95 Å was collected from a single crystal at 100 K. Raw data images were processed and scaled with XDS[41], and Aimless[42] using the CCP4 suite 7.0 (CCP4, 1994). Initial phases for structure solution were obtained by molecular replacement using the MoRDa pipeline[43]. The working models were refined using Refmac5[44] and manually adjusted with Coot[45]. Validation was performed with Molprobity[46], with 95.5% of residues within the favored Ramachandran region, and no outliers. Figures were drawn with PyMOL (Schrödinger, LLC, New York, USA).

**Reporting summary**. Further information on research design is available in the Nature Research Reporting Summary linked to this article.

## Data availability

The atomic coordinates and structure factors (code 6HOX) have been deposited in the Protein Data Bank. Accession numbers of CP032452 for *Pbm* genome and CP032455 for *pPbmMP*, and CP032453 and CP032454 for two other plasmid DNA sequences are available at the NCBI database (GenBank). The *Pbp* and *Pb* whole-genome shotgun projects have been deposited at DDBJ/ENA/GenBank under the accession numbers RANB00000000 and QZNC00000000, respectively. The source data underlying Figs. 2b–f, 3a, and 4c are provided as a Source Data file.

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

## Acknowledgements

We thank S. Pan from the University of California Proteomic Facility for his support in mass spectrometry analysis (supported by a NIH shared instrumentation grant (S10 OD010669)) and scientists at station ID30B of ESRF (Grenoble, France) for their support during crystallography data collection. We thank Dr. Daniel Lundin for help with the phylogenetic network. The research was funded in part through grants from the National Institutes of Health, 1 R01AI123390 and 1R21AI070873, and the University of California Agricultural Experiment Station CA-R-NEU-7395-H to S.S.G. and by the Swedish Research Council (2014–5667) and the Swedish Cancer Society to P.S.

## Author contributions

S.S.G. conceived the overall project from mosquitocidal strains isolated by H.L.L. With the exception of the following, all experiments were performed by E. C. N.Q. created the Pbm∆109 mutant, S.C. analyzed *Pbm*, *Pbp*, and *Pb* genomes, J.C. created the NTNH-OrfX1-3-PMP1 construct, and H.S.D. determined its LC50, S.S.G. did the mice injections,

G.M. crystallized the HC of PMP1 and determined the structure, P.S. created the phylogenetic network and carried out structural and bioinformatical analysis. E.C. and S.S.G. wrote and revised the manuscript with contributions from G.M. and P.S. and G.M. and P.S. helped revise the manuscript.

## Additional information

**Competing interests:** S.S.G., E.C., and J.C. have filed a provisional patent application. The other authors declare no competing interests.

