## [Peer Review File · Nature Communications]

Reviewers' Comments:

Reviewer #1:

Remarks to the Author:

Botulinum neurotoxins (BoNTs) show a wide diversity. They are divided into more than 40 types and subtypes and are produced by various bacteria mainly from the *Clostridium* genus. Recent works revealed novel BoNT types and novel non-clostridium BoNT producer microorganisms. Until now, BoNTs have been found to target only vertebrate animals. Here, the manuscript of Contreras et al. reports a novel BoNT type able to specifically target certain insects (anopheles mosquitos) that is produced by some *Clostridium bifermentans* strains already known to synthesize an insecticidal toxin from the Cry family. The work is well performed and opens novel perspectives in insect control and shed some light on the evolution and dissemination of neurotoxin genes.

L103-105 The lethality expressed as % mortality is not clear. The determination of LD50 would be more precise.

L106-107. a reference is required

L107. It is not clear how OrfXs are involved in toxicity. Regarding the BoNT protection against gut degradation, NTNH is apparently enough. OrfXs have not been found to have a protective effect.

L146-147. PMP1 is shown to cleave mosquito syntaxin in vitro. BoNT/C has been found to cleave mammal syntaxin in vitro and also SNAP25 in neuronal or chromaffin cells. Does PMP1 cleave additional SNARE in vivo using mosquito neuronal cells?

L194-204. Toxin mutants on the putative ganglioside binding site have been tested by toxicity on insect larvae. This would be nicely improved by investigation of toxin binding to gangliosides, by ELISA for example.

Reviewer #2:

Remarks to the Author:

This is a well conceived, and conducted study reporting the isolation of a new clostridial-like neurotoxin from *Paraclostridium bifermentans* named PMP1 with specificity for anopheline mosquitoes. There are several remarkable features reported in this Communication that deserve its publication. However, there are aspects of the study that could be improved to make it more compelling and that could be completed within a realistic time frame without introducing a major delay in publication.

The comparative genomic analysis led the authors to identify PMP1 with high similarity to botulinum and tetanus neurotoxins both in sequence as well as in overall structural architecture in terms of functional motifs. The toxicity correlated with the presence of a metalloprotease that selectively cleaved mosquito syntaxin and not human or mice syntaxin. The cleavage assay using the recombinant SNARE proteins as well as the syntaxin mutations are consistent with the interpretation. And, the fact that the PMP1 is toxic to mosquito larvae and not to mice is significant. The video showing the data on the mosquito larvae and the lack of activity of the E209Q mutant in the active site of the metalloprotease strengthen the notion that the protease activity accounts for the ultimate toxicity of PMP1. The structural analysis of the HCC domain indicates that a new SWYG motif in loop 1 is a determinant of toxicity and, in this respect, differs from other clostridial neurotoxins.

Importantly, as far as vector control is concerned, this report is intriguing and tantalizing. The discussion of this aspect deserves more elaboration.

The gel assay in Figure 3b should be improved.

Figure 5A (line186) was not included. It may be a typo for Figure 4A.

Given the expertise and capabilities of the Swedish group an effort to determine the structure of the recombinant light chain in complex with syntaxin is within reach. As it stands, the mutation analysis for the inactivity of the protease to cleave human syntaxin despite of the conservation of the cleavage site is suggestive of the mechanism presented and consistent with current knowledge on the requirements of exosites; however, it fails to be definitive. This addition to the report would be a major improvement in a revised version of the manuscript.

Reviewer #3:

Remarks to the Author:

This is a very interesting work, excellent and very comprehensive science, as well as very useful and timely outcomes, in terms of practical solutions for mosquito control. The study is technically well executed and the manuscript concise and very well written, appropriate for a broad audience. I believe the article should be accepted for publication and have a few minor comments only, that the authors may wish to consider in their revised version:

- 1) How does the toxicity of the injected proteins compare to other neurotoxins (spider or scorpion) or small molecule insecticides. Is the novelty of this compound the fact that it can cross the gut as a neurotoxic complex or does it better kill insects once inside the hemolymph. It would seem to be the latter, but if not then it would be interesting to consider other proteins in the complex (NTNH, ORFX1-3) as a more general gut delivery method.
- 2) How does the toxicity Bt lines expressing various components of the PMP cassette compare with the wild type bacteria? In other words what is the contribution of the PMP cassette to toxicity in vivo. In figure 2 B the authors have not included a positive control that is the wild type bacteria (containing potential toxicity factors such as Cry16 and Cry17). The reason this might be important is because, perhaps the Cry toxins are synergizing with the other components of the system to increase gut penetration (not likely because these don't appear to work in Anopheles) but would still be nice.
- 3) A few of their figures have spell check lines under them.

Response to referees

We thank the reviewers for their positive response and constructive suggestions. We believe the resulting changes have improved our manuscript.

Reviewers' comments:

Reviewer #1 (Remarks to the Author):

Botulinum neurotoxins (BoNTs) show a wide diversity. They are divided into more than 40 types and subtypes and are produced by various bacteria mainly from the Clostridium genus. Recent works revealed novel BoNT types and novel non-clostridium BoNT producer microorganisms. Until now, BoNTs have been found to target only vertebrate animals. Here, the manuscript of Contreras et al. reports a novel BoNT type able to specifically target certain insects (anopheles mosquitos) that is produced by some Clostridium bifermentans strains already known to synthesize an insecticidal toxin from the Cry family. The work is well performed and opens novel perspectives in insect control and shed some light on the evolution and dissemination of neurotoxin genes.

We thank the reviewer for the positive comments

L103-105 The lethality expressed as % mortality is not clear. The determination of LD50 would be more precise.

In mice a precise dose can be given through gavage to obtain LD50 values. However, the amount of toxin ingested by mosquito larvae in bioassays (feeding experiments) cannot be determined. To obtain LD50 values for the PMP1 toxin we did mosquito injections and this data is presented in Fig 2C.

For this reason it is more usual to obtain LC50 values for mosquito bioassays. We agree with this reviewer that obtaining the LC50 values for all the constructs would be best. Unfortunately the efficiency of our expression system does not enable us to sufficiently concentrate Bt cultures containing the constructs to reach mortality values for the calculation of LC50. Hence only the construct with the five genes gives a reliable LC50. We have added this value to the Figure 2 legend (line 619-620). Because of these constraints we believe for comparison, the toxicity of equal amounts of bacterial cultures is the best available approach. We continue to work on more efficient expression systems to get mortality data that allows calculation of LC50 values and also evaluate the role of individual OrfXs.

L106-107. a reference is required

A reference is added.

L107. It is not clear how OrfXs are involved in toxicity. Regarding the BoNT protection against gut degradation, NTNH is apparently enough. OrfXs have not been found to have a protective effect.

The reviewer is correct that OrfXs have not been found to have a protective effect in mammalian toxicity. Our data does not imply that there is a protective role in mosquitoes, but rather the pmp1-orfX1-3-ntnh construct has higher toxicity than in the absence of these genes. How this is manifested is unclear at present and is a focus of our current investigation. We had hypothesized they help in increasing toxin binding. But we do not have any evidence to date that this occurs. Hence this is a project that will require extensive work.

We have changed the sentences to differentiate the role of NTNH, since as in mammals, it is likely sufficient for protection of the PMP1 protein and have added some discussion on the potential role of OrfX's. (Lines 227-232)

L146-147. PMP1 is shown to cleave mosquito syntaxin in vitro. BoNT/C has been found to cleave mammal syntaxin in vitro and also SNAP25 in neuronal or chromaffin cells. Does PMP1 cleave additional SNARE in vivo using mosquito neuronal cells?

We thank the reviewer for this suggestion. We really wish we could do these experiments. Unfortunately there are no mosquito neuronal cells in culture, and the use of primary neurons is currently technically impossible since we cannot get sufficient neurons. Instead we are planning to use vertebrate cells to analyze in vivo cleavage but this has to be done with fused toxins that use PMP-LC with a vertebrate binding domain to facilitate cell entry. However, data from such experiments would only inform us on cross-SNARE specificity in vertebrates, since a few changes between mosquito and human SNAREs can make a difference in PMP-LC ability to cleave, as observed with syntaxin1.

L194-204. Toxin mutants on the putative ganglioside binding site have been tested by toxicity on insect larvae. This would be nicely improved by investigation of toxin binding to gangliosides, by ELISA for example.

We thank the reviewer for this suggestion. Indeed we did these experiments. Preliminary experiments indicated the binding domain had more selectivity to GM1. However, GM1 is not present in insects, since although dipterans have complex glycosphingolipids with sialic acids there is no evidence that mosquitoes have gangliosides. Further, GM1 shows low affinity to the binding domain ($\sim 10^{-5}M$) in SPR experiments, and in preliminary results GM1 does not compete with the PMP1 binding to mosquito head membrane preparations. We followed up these experiments with binding of the PMP1 binding domain to a glycan array (ncfg.hms.harvard.edu/). However, none of the glycans in the array showed any significant binding.

Reviewer #2 (Remarks to the Author):

This is a well conceived, and conducted study reporting the isolation of a new clostridial-like neurotoxin from *Paraclostridium bifermentans* named PMP1 with specificity for anopheline mosquitoes. There are several remarkable features reported in this Communication that deserve its publication. However, there are aspects of the study that could be improved to make it more compelling and that could be completed within a realistic time frame without introducing a major delay in publication.

The comparative genomic analysis led the authors to identify PMP1 with high similarity to botulinum and tetanus neurotoxins both in sequence as well as in overall structural architecture in terms of functional motifs. The toxicity correlated with the presence of a metalloprotease that selectively cleaved mosquito syntaxin and not human or mice syntaxin. The cleavage assay using the recombinant SNARE proteins as well as the syntaxin mutations are consistent with the interpretation. And, the fact that the PMP1 is toxic to mosquito larvae and not to mice is significant. The video showing the data on the mosquito larvae and the lack of activity of the E209Q mutant in the active site of the metalloprotease strengthen the notion that the protease activity accounts for the ultimate toxicity of PMP1. The structural analysis of the HCC domain indicates that a new SWYG motif in loop 1 is a determinant of toxicity and, in this respect, differs from other clostridial neurotoxins.

Importantly, as far as vector control is concerned, this report is intriguing and tantalizing. The discussion of this aspect deserves more elaboration.

We thank the reviewer for the positive comments. We did not want to speculate, although we have thought of this. But on the reviewers' suggestion we have added to the discussion (Lines 271-276)

The gel assay in Figure 3b should be improved.

The gel identifies the cleaved peptide, and since this is very small in size in SDS-PAGE gels it is poorly stained by Coomassie, and a high amount of total protein has to be loaded. Hence the non-cleaved and N-terminus products appear overstained. We tried a number of times to improve the detection of the 4kDa band but unfortunately all trials had similar reproduceable appearance. Our focus was to identify the 4kDa band, which we did and subjected to LC/MS/MS analysis for confirmation.

Figure 5A (line186) was not included. It may be a typo for Figure 4A.

We agree. It should read Figure 4B and the typo is corrected. (Line 189)

Given the expertise and capabilities of the Swedish group an effort to determine the structure of the recombinant light chain in complex with syntaxin is within reach. As it stands, the mutation analysis for the inactivity of the protease to cleave human syntaxin despite of the conservation of the cleavage site is suggestive of the mechanism presented and consistent with current knowledge on the requirements of exosites; however, it fails to be definitive. This addition to the report would be a major improvement in a revised version of the manuscript.

We thank the reviewer for this suggestion and the reviewer's confidence. We concur that the crystal structure of the light chain, and of its complex with syntaxin, would greatly help to elucidate the substrate binding mechanism. We successfully produced recombinant material and have been actively trying to obtain crystals of the light chain, including co-crystallization trials with syntaxin for the last year. However, despite our best efforts and thousands of conditions tested, we have not been able to obtain any crystals to date. This work is, however, still on going, including since the review.

Reviewer #3 (Remarks to the Author):

This is a very interesting work, excellent and very comprehensive science, as well as very useful and timely outcomes, in terms of practical solutions for mosquito control. The study is technically well executed and the manuscript concise and very well written, appropriate for a broad audience. I believe the article should be accepted for publication and have a few minor comments only, that the authors may wish to consider in their revised version:

1) How does the toxicity of the injected proteins compare to other neurotoxins (spider or scorpion) or small molecule insecticides. Is the novelty of this compound the fact that it can cross the gut as a neurotoxic complex or does is it able to better kill insects once inside the hemolymph. It would seem to be the latter, but if not then it would be interesting to consider other proteins in the complex (NTNH, ORFX1-3) as a more general gut delivery method.

We thank the reviewer for the positive comments and suggestions.

Based on published literature on spider or scorpion toxins, PMP1 has at 10-100x higher toxicity, if not greater, when injected. This is now indicated on lines 121-122. In large part this is because Pmp1 is an enzyme, while the spider or scorpion toxins are ligands of specific receptors or channels. However, comparisons with small molecule insecticides are difficult since they are usually applied topically.

We agree with the reviewer that the novelty of the finding is that there is very high selectivity. But once in the hemolymph there is less selectivity. We have thought of the mechanism of cell entry quite a bit, but it is quite unlikely that any protein could be used by NTNH, ORFX1-3 as a delivery mechanism based on what is known about the role of NTNH in mammalian systems. The reviewer has come with a very interesting suggestion and worth exploring and needs experimental work. But being speculative we have not added it to the discussion.

2) How does the toxicity Bt lines expressing various components of the PMP cassette compare with the wild type bacteria? In other words what is the contribution of the PMP cassette to toxicity in vivo. In figure 2B the authors have not included a positive control that is the wild type bacteria (containing potential toxicity factors such as Cry16 and Cry17). The reason this might be important is because, perhaps the Cry toxins are synergizing with the other components of the system to increase gut penetration (not likely because these don't appear to work in Anopheles) but would still be nice.

We thank the reviewer for the enquiry. Data for the wild type bacteria is presented in Supplementary Data (Table 1) and that of the Bt construct expressing the full PMP cassette in Figure 1B. It is clear that the wild-type is more toxic than the full PMP cassette. At this stage we cannot exclude the contribution of other proteins from the megaplasmid. However, addition of the MPP or P47 proteins to the full PMP cassette does not change the toxicity. Further, in preliminary results we do not observe synergy between Cry operon, which is not toxic to Anopheles, and the toxicity of the pmp operon.

We believe that since Paraclostridium and Bacillus have differences in lysis with time, toxins expressed in Bacillus are not fully available to mosquito larvae and hence incomplete lysis of B. thuringiensis leads to a lower uptake of toxin by the larvae. (see Line 113-116).

3) A few of their figures have spell check lines under them.

Spell check lines were removed

Reviewers' Comments:

Reviewer #1:

Remarks to the Author:

Botulinum neurotoxins (BoNTs) are extremely potent toxins that show a wide diversity regarding their amino acid composition, cell surface receptors, and intracellular targets. However, they retain a common structure and mode of action consisting of the blockade of neurotransmitter at target neuronal cells. Thereby, BoNTs are divided into more than 40 types and subtypes and are produced by various bacteria mainly from the Clostridium genus. Recent works revealed novel BoNT types and novel non-clostridium BoNT producer microorganisms. Until now, BoNTs have been found to induce flaccid paralysis only in vertebrate animals. Here, the manuscript of Contreras et al. reports a novel BoNT type able to specifically target certain insects (anopheles, mosquitos) that is produced by some Clostridium bifermentans strains already known to synthesize an insecticidal toxin from the Cry (Crystal) family. This work opens novel perspectives in insect control and shed novel light on the evolution and dissemination of botulinum neurotoxin genes.

Reviewer #2:

Remarks to the Author:

The revised version has improved significantly and merits publication essentially as is.

Reviewer #3:

Remarks to the Author:

The authors have made sufficient amendments and provided appropriate explanations, to cover all points raised by the reviewers and further improve their manuscript. Thus, their work should be accepted for publication in Nature Communication, in my opinion. It is a very good paper, with significant impact in the field.

Response to reviewers

Reviewer #1 (Remarks to the Author):

Botulinum neurotoxins (BoNTs) are extremely potent toxins that show a wide diversity regarding their amino acid composition, cell surface receptors, and intracellular targets. However, they retain a common structure and mode of action consisting of the blockade of neurotransmitter at target neuronal cells. Thereby, BoNTs are divided into more than 40 types and subtypes and are produced by various bacteria mainly from the Clostridium genus. Recent works revealed novel BoNT types and novel non-clostridium BoNT producer microorganisms. Until now, BoNTs have been found to induce flaccid paralysis only in vertebrate animals. Here, the manuscript of Contreras et al. reports a novel BoNT type able to specifically target certain insects (anopheles, mosquitos) that is produced by some Clostridium bifermentans strains already known to synthesize an insecticidal toxin from the Cry (Crystal) family. This work opens novel perspectives in insect control and shed novel light on the evolution and dissemination of botulinum neurotoxin genes.

We concur with the assessment of this reviewer's paragraph, and have nothing to add.

Reviewer #2 (Remarks to the Author):

The revised version has improved significantly and merits publication essentially as is.

We concur with the assessment of this reviewer's paragraph, and have nothing to add.

Reviewer #3 (Remarks to the Author):

The authors have made sufficient amendments and provided appropriate explanations, to cover all points raised by the reviewers and further improve their manuscript. Thus, their work should be accepted for publication in Nature Communication, in my opinion. It is a very good paper, with significant impact in the field.

We concur with the assessment of this reviewer's paragraph, and have nothing to add.